# Mutational Profile Enables the Identification of a High-Risk Subgroup in Myelodysplastic Syndromes with Isolated Trisomy 8

**DOI:** 10.3390/cancers15153822

**Published:** 2023-07-27

**Authors:** Sofía Toribio-Castelló, Sandra Castaño, Ángela Villaverde-Ramiro, Esperanza Such, Montserrat Arnán, Francesc Solé, Marina Díaz-Beyá, María Díez-Campelo, Mónica del Rey, Teresa González, Jesús María Hernández-Rivas

**Affiliations:** 1IBSAL, IBMCC, CSIC, Cancer Research Center, University of Salamanca, 37007 Salamanca, Spain; storibio@usal.es (S.T.-C.); jmhr@usal.es (J.M.H.-R.); 2Department of Hematology, University Hospital of Salamanca, 37007 Salamanca, Spain; mdiezcampelo@usal.es (M.D.-C.); tergonma@usal.es (T.G.); 3CIBERONC, Research Group CB16/12/00233, 37007 Salamanca, Spain; 4Hematology Department, Hospital Clínic de Barcelona, August Pi i Sunyer Biomedical Research Institute (IDIBAPS), University of Barcelona (UB), 08007 Barcelona, Spain; 5Hematology Department, Hospital La Fe, 46026 Valencia, Spain; 6Hematology Department, Catalan Institute of Oncology (ICO)-Hospital Duran i Reynals, 08908 L’Hospitalet de Llobregat, Spain; 7MDS Group, Institut de Recerca Contra la Leucèmia Josep Carreras, ICO-Hospital Germans Trias i Pujol, Universitat Autònoma de Barcelona, 08193 Badalona, Spain

**Keywords:** myelodysplastic syndromes, NGS, isolated trisomy 8, prognostic stratification, AML progression

## Abstract

**Simple Summary:**

Trisomy 8 (+8) is one of the most frequent cytogenetic alterations found in myelodysplastic syndromes (MDS). MDS patients harboring isolated +8 show clinical heterogeneity, and life expectancy varies between several months and several years after diagnosis. We aimed to investigate whether the mutational profile of isolated +8 MDS patients could help to clarify the heterogeneous prognosis of these patients. In fact, we found that mutations in *STAG2*, *SRSF2* and *RUNX1* are independent prognostic factors, enough to define the course of the disease in terms of overall survival and leukemic transformation in isolated +8 MDS. Therefore, these findings might help to identify patients at a high risk of evolving disease and open new horizons by changes in the management of a high subset of patients within MDS with isolated +8.

**Abstract:**

Trisomy 8 (+8) is the most frequent trisomy in myelodysplastic syndromes (MDS) and is associated with clinical heterogeneity and intermediate cytogenetic risk when found in isolation. The presence of gene mutations in this group of patients and the prognostic significance has not been extensively analyzed. Targeted deep sequencing was performed in a cohort of 79 MDS patients showing isolated +8. The most frequently mutated genes were: *TET2* (38%), *STAG2* (34.2%), *SRSF2* (29.1%) and *RUNX1* (26.6%). The mutational profile identified a high-risk subgroup with mutations in *STAG2*, *SRSF2* and/or *RUNX1*, resulting in shorter time to acute myeloid leukemia progression (14 months while not reached in patients without these mutations, *p* < 0.0001) and shorter overall survival (23.7 vs. 46.3 months, *p* = 0.001). Multivariate analyses revealed the presence of mutations in these genes as an independent prognostic factor in MDS showing +8 isolated (HR: 3.1; *p* < 0.01). Moreover, 39.5% and 15.4% of patients classified as low/intermediate risk by the IPSS-R and IPSS-M, respectively, were re-stratified as a high-risk subgroup based on the mutational status of *STAG2*, *SRSF2* and *RUNX1*. Results were validated in an external cohort (*n* = 2494). In summary, this study validates the prognosis significance of somatic mutations shown in IPSS-M and adds *STAG2* as an important mutated gene to consider in this specific subgroup of patients. The mutational profile in isolated +8 MDS patients could, therefore, offer new insights for the correct management of patients with a higher risk of leukemic transformation.

## 1. Introduction

Myelodysplastic syndromes (MDSs) are a group of clonal hematological disorders characterized by clinical heterogeneity and genetic diversity [1,2]. MDSs display peripheral cytopenias due to ineffective hematopoiesis, morphologic dysplasia in one or more cell lineages and an increased risk of leukemia transformation. In fact, more than 30% of MDS patients progress to acute myeloid leukemia (AML). Approximately, 50% of patients show cytogenetic alterations and more than 90% have at least one genetic lesion at diagnosis [2,3,4,5].

In the last decade, the identification of genetic alterations by next-generation sequencing (NGS) has changed the understanding of MDS pathogenesis [1,6]. In fact, in 2017 the presence of somatic mutations was incorporated for the first time as a new parameter for a better diagnosis of morphological subtypes [3]. Moreover, the recently published update of the diagnosis classification by both the World Health Organization (WHO) and the International Consensus Classification (ICC), defines the presence of genetic abnormalities as an important feature to consider in diagnosis of myeloid malignancies [7,8].

Until now, the Revised International Prognostic Scoring System (IPSS-R) was the most commonly used tool for prognostic assessment of primary untreated MDSs in adults. Cytogenetic alterations are one of the most important features that have an impact on risk-stratification [9]. In fact, a cytogenetic IPSS scoring system was implemented to determine the risk of each karyotypic alteration in order to incorporate this molecular information into the IPSS-R [10]. In addition, a role for somatic mutations in the outcome of MDS patients has been described and genetic variants have been incorporated into the recently published International Molecular Prognostic Scoring System (IPSS-M). The classification of MDS patients by the IPSS-M shows improved discrimination between risk subgroups by including information from alterations in 31 recurrently mutated genes in MDSs [11].

One of the most common cytogenetic alterations found in MDSs is trisomy 8 (+8), which is present in 10% of patients with chromosomal abnormalities [12,13]. Patients with isolated +8 have an intermediate cytogenetic risk and are also characterized by clinical heterogeneity: life expectancy is different among patients, varying between several months and several years after diagnosis. Two subgroups of MDS patients with isolated +8 have been described, showing differences in overall survival (OS) based on myeloproliferative features [14]. Some of these patients showed a higher number of mutations. In that way, some studies have described the presence of mutated genes in MDS displaying +8, such as *ASXL1*, *EZH2*, *STAG2*, *SRSF2* or *U2AF1* [14,15,16,17]. Furthermore, only one study described a group of genes worsening the prognosis of MDS patients, but those cases were associated with myeloproliferative features [14]. Thus, the role of somatic mutations in isolated +8 MDS is still undefined, as there is no evidence of its impact in the outcome of isolated +8 in large cohorts of MDS patients without features such as myeloproliferation.

Given the clinical heterogeneity of MDS patients with isolated +8, accurate survival prediction and risk stratification are critical for the proper management of and treatment decision-making for these patients. Therefore, the aim of the study herein was to evaluate the mutational landscape of isolated +8 and to determine whether specific somatic mutations might help to identify patients with adverse outcomes in this heterogeneous group of MDS.

## 2. Patients and Methods

### 2.1. Patients Cohort and Validation Cohort

A total of 2602 patients with de novo MDS, diagnosed between 1997 and 2022, were analyzed for incidence of trisomy 8. There were 94 patients (3.6%) who showed isolated +8, while 30 cases (1.1%) showed +8 in combination with additional cytogenetic alterations. Most of these combinations (80%) were complex karyotypes (CKs) and, consequently, were not included in the study. Clinical and biological data are shown in Table 1. Patients with isolated +8 were compared to IPSS-R-based matched MDS controls without +8 in a 3:1 ratio (282 patients). NGS data were available in a total of 187 patients: 79 out of 94 MDS patients with isolated +8 and in 108 out of the 282 control patients. The procedures followed were in accordance with the ethical standards of the Institutional Committee on Human Experimentation and with the Helsinki Declaration of 1975. The research was approved by the Local Ethics Committee from the University Hospital of Salamanca, Spain.

In addition, an external cohort comprising 2494 MDS patients from the International Working Group for the Prognosis of MDS was also analyzed to validate the results [11]. A total of 94 patients showing isolated +8 were compared to 2311 MDS cases as controls without +8, while 89 cases were excluded due to the presence of additional cytogenetic alterations together with +8 (*n* = 81) or because the status of chromosome 8 could not be ascertained (*n* = 8).

### 2.2. Sample Procedure and DNA Extraction

Bone marrow (BM) and peripheral blood (PB) samples were obtained from patients at diagnosis. Mononuclear cells were isolated by gradient density using Ficoll. 

Genomic DNA was obtained using QIAamp DNA Mini Kit (Qiagen, Hilden, Germany) following manufacturer’s standard protocol. The concentration of the extracted DNA was assessed by Qubit 2.0 Fluorometer system (Life Technologies, Carlsbad, CA, USA) and the integrity was analyzed using a TapeStation 4200 (Agilent Technologies, Santa Clara, CA, USA) and a Nanodrop spectrophotometer (ND-1000, NanoDrop Technologies, Wilmington, DE, USA) by measuring absorbance ratio at 260/230 and 260/280 nm.

### 2.3. Cytogenetics and FISH

Conventional G-banding analysis (CBA) data were available for all patients, and karyotypes were described in accordance with the International System for Human Cytogenetic Nomenclature [18].

Karyotype information was confirmed in all patients by interphase fluorescence in situ hybridization (FISH).

### 2.4. Next-Generation Sequencing

Samples from 187 patients underwent targeted deep sequencing using an update of a previously validated in-house custom capture-enrichment panel [19] (SureSelect XT HS Target Enrichment System, Agilent Technologies, Santa Clara, CA, USA) of 92 genes related to the pathogenesis of myeloid malignancies (Appendix A). Sequencing libraries were prepared according to manufacturer’s instructions and sequenced on Illumina MiSeq or NexSeq 500 sequencers.

The mean coverage of TDS was 665× (range 251–1198) and 99.5% of target regions were captured at a level higher than 100×.

For true oncogenic somatic variant calling, a severe criterion for variant filtering was applied. Variants were considered candidate somatic mutations based on the following criteria: (i) variants with ≥10 mutated reads; (ii) described in COSMIC and/or ClinVar as being cancer-associated and known hotspot mutations; and (iii) classified as deleterious and/or probably damaging by PolyPhen-2 and SIFT web-based platforms, as previously described [19].

### 2.5. Statistical Analysis

Dichotomous variables were compared between different groups using the χ^2^ test and continuous variables by the Student’s *t*-test, Mann–Whitney and Kruskal–Wallis tests. Results were considered significant at *p* < 0.05.

Multivariate Cox regression analysis was applied to compare clinical and molecular characteristics of patients. Survival and disease progression analyses were performed with Kaplan–Meier method and groups were compared using two-sided log-rank test. OS was defined as the period from the date of initial diagnosis to the date of death regardless of the cause. Data were censored at the last follow-up in (a) patients with loss of follow-up and (b) patients undergoing hematopoietic stem cell transplantation (HSCT), with date of HSCT as the last point of time considered. Time to AML progression were considered as the time from the MDS to the secondary AML diagnoses and censored in patients without transformation at last follow-up or date of death.

Statistical analyses were performed using SPSS Statistics version 25 (IBM Corporation, Amonk, NY, USA), Graphpad prism version 5.03 for Windows, (Graphpad software, San Diego, CA, USA) and R version 4.0.2 with biostatistical packages (https://cran.r-project.org/ accesed on 25 May 2022).

## 3. Results

### 3.1. MDS with Isolated Trisomy 8 Have a Differential Mutational Pattern

A total of 289 oncogenic mutations were identified in 43 genes in isolated +8 MDS patients. *TET2* was the most frequently mutated gene in isolated +8 MDS patients (38%). Interestingly, *STAG2* (34.2%), *SRSF2* (29.1%) and *RUNX1* (26.6%) were also frequently mutated in MDS patients displaying isolated +8. Nine genes were mutated in more than 10% of patients, with an additional ten genes carrying mutations in 5 to 10% of patients (Figure 1a; Appendix A). The mean number of mutations per patient was 3.7 (0–10). Seventy-one (89.8%) patients had at least one mutated gene: nine patients (11.4%) had only one mutated gene while nine patients (11.4%) showed two mutated genes, and the remaining fifty-three cases (67.1%) had three or more mutated genes.

To investigate whether the pattern of recurrently mutated genes was different in this set of patients, mutational profiles were compared between isolated +8 MDS patients and the 282 IPSS-R-matched MDS patients without +8, defined as the control group. A differential mutational pattern on isolated +8 patients was found compared to the controls (Figure 1b). Thus, the incidence of mutations in *STAG2*, *SRSF2*, *RUNX1*, *EZH2*, *ASXL1*, *ZRSR2* and *IDH2* was higher in the isolated +8 MDS group than in the control cohort (*p* < 0.05). By contrast, mutations in *SF3B1* and *DNMT3A* were more frequent in non-trisomy 8 MDS group (*p* < 0.001). In addition, the median number of mutations per patient was higher in isolated +8 patients compared to those MDS patients without +8 (4 vs. 3, *p* < 0.001; Figure 1c).

### 3.2. Mutational Profile Allows the Distinction of a Worse Prognosis High-Risk-Like Subgroup in MDS with Isolated Trisomy 8

The presence of mutations in *STAG2*, *SRSF2* and *RUNX1* in isolated +8 MDS patients was significantly associated with shorter time to AML (*p* < 0.05; Figure 2a, Appendix A). Mutations in either *STAG2*, *SRSF2* and/or *RUNX1* were, therefore, considered as an entire group of alterations worsening the prognosis among patients with MDS and isolated +8 (Appendix A). In fact, isolated +8 MDS with mutations in at least one of these three genes, showed a median time to AML progression and OS similar to high/very high-risk patients from the control group without +8 stratified by the IPSS-R (AML: 14 vs. 11.4 months; OS: 23.7 vs. 10.1 months, respectively; *p* > 0.05; Figure 2b,c). By contrast, isolated +8 MDS patients without mutations in *STAG2*, *SRSF2* and *RUNX1* showed a longer time to AML progression (comparable to very low/low-risk MDS patients without +8 classified by IPSS-R) and remained as MDS patients with a low–intermediate IPSS-R risk in terms of OS (Figure 2b,c).

Moreover, time to AML progression in isolated +8 patients with mutations in *STAG2*, *SRSF2* and/or *RUNX1* was similar to CK (14 vs. 11.7 months; *p* > 0.05; Appendix A) and tended to have a comparable OS to those patients (Appendix A). In addition, time to AML progression in isolated +8 with no mutations in these genes was even longer than in MDS control patients with good cytogenetic risk (median not reached vs. 29.3 months; *p* < 0.05; Appendix A) and a similar OS was observed between them (46.3 vs. 34.9 months, respectively; *p* > 0.05; Appendix A).

Clinical features of patients with isolated +8 and the presence of mutations in *STAG2*, *SRSF2* or *RUNX1* differed from MDS patients without +8 classified as high/very high risk by the IPSS-R in levels of hemoglobin and percentage of BM blasts (Appendix A). Interestingly, despite having higher hemoglobin levels and lower counts of blast in BM, no differences were found in patient’s outcome in terms of both AML progression and OS (Figure 2b,c). Otherwise, patients with isolated +8 and without mutations in neither *STAG2*, *SRSF2* nor *RUNX1*, showed a similar age, hemoglobin level, absolute neutrophils count and outcome to MDS patients without +8, classified as low risk by the IPSS-R, but a higher percentage of BM blasts and lower levels of platelets and ring sideroblasts (Figure 2b,c, Appendix A).

### 3.3. Mutational Status of STAG2, SRSF2 and/or RUNX1 Is an Independent Prognostic Factor Associated with Shorter Time to Progression to AML and Overall Survival in MDS with Isolated Trisomy 8

In a multivariate Cox regression analysis, clinical parameters included in the IPSS-R were separately evaluated. In this regard, age (≥75 years), percentage of BM blasts (≥5%), number of cytopenias (>2) and the mutational status of *STAG2*, *SRSF2* or *RUNX1* were considered. Alterations in these genes were associated with shorter time to AML (*p* < 0.01; Table 2). Interestingly, the number of mutations per patient was not associated with time to AML (*p* > 0.05; Appendix A), indicating that mutations in either *STAG2*, *SRSF2* and/or *RUNX1* are enough to predict early disease progression in isolated +8 MDS patients.

OS was also evaluated in a similar multivariate analysis considering age (≥75 years), percentage of BM blasts (≥5%), number of cytopenias (>2) and the mutational status of *STAG2*, *SRSF2* and *RUNX1*. Only mutations in these genes remained as independent predictors of worse outcomes (*p* = 0.01; Table 2). By contrast, the number of mutations per patient (>4) was not associated with shorter OS (Appendix A). Furthermore, multivariate analyses also revealed that when considering a percentage of BM blasts higher than 10%, the mutational status of *STAG2, SRSF2* and *RUNX1* was still an independent prognostic factor of shorter OS, together with >10% blast counts and age (Appendix A).

### 3.4. The Newly Defined Prognostic Mutations Could Refine Prognosis of Patients with Isolated Trisomy 8 within the IPSS-R and the IPSS-M Systems

Patients with MDS and isolated +8 were classified within both IPSS-R and IPSS-M. A comparison between our defined mutational prognostic signature and both the IPSS-R and IPSS-M was assessed to evaluate whether the mutational status of *STAG2*, *SRSF2* and *RUNX1* could add value to these prognostic scoring systems in isolated +8 MDS. Remarkably, 39.5% of patients categorized as low/intermediate risk by the IPSS-R showed a high-risk mutational profile (this is, mutations in either *STAG2*, *SRSF2* or *RUNX1*) (Figure 3).

Otherwise, 15.4% of low/moderate–low-risk patients as categorized by the IPSS-M were classified as high risk by the mutational profile (Figure 3).

As a result, most of the patients showing mutations in *STAG2*, *SRSF2* and/or *RUNX1* were categorized as moderate–high, high or very high risk by the IPSS-M (88.9%) while only 51.6% were categorized as high or very high by the IPSS-R. Consequently, the risk of isolated +8 patients with mutations in *STAG2*, *SRSF2* and/or *RUNX1* was refined in a total of 11.1% of patients by the IPSS-M and in 48.4% of cases by the IPSS-R (Appendix A).

### 3.5. STAG2, SRSF2 and RUNX1 Differentiate a Subgroup of Isolated Trisomy 8 MDS with Worse Outcome in an Independent Validation Cohort 

Finally, to further characterize the mutational landscape associated with isolated +8 in MDS, additional analyses were carried out in an external validation cohort. Thus, a previously published MDS population comprising 2494 patients from the International Working Group for the Prognosis of MDS was evaluated [11].

A total of 94 MDS patients presented isolated +8 (3.8%) and clinical characteristics were similar to those of the discovery cohort (Appendix A). In this set of patients, the most frequently mutated genes were *ASXL1* (44.7%), *TET2* (39.4%) and *STAG2* (36.2%). The mutational incidence observed in most of the genes were similar to those of our cohort of MDS with isolated +8 patients (Appendix A). Twenty-three (24.5%) and twenty-one (22.3%) patients showed mutations in *SRSF2* and *RUNX1*, respectively. In addition, the median number of mutations per patient was higher in isolated +8 patients compared to MDS without +8 (four vs. three, respectively; *p* < 0.0001), as in our discovery cohort.

As in our own cohort, the presence of mutations in *STAG2*, *SRSF2* and/or *RUNX1* was able to distinguish a worse prognosis subgroup of patients. In fact, patients with mutations in these genes showed a similar time to AML and OS to high/very high-risk IPSS-R MDS patients (2.2 years of AML transformation in the high-risk patients without +8 and 1.6 and 1.2 years of OS, respectively; *p* > 0.05; Appendix A). Multivariate analysis also revealed an independent prognostic value for these mutations in isolated +8 MDS (*p* < 0.05; Appendix A).

## 4. Discussion

MDS patients are characterized by clinical and genetic heterogeneity that generally translate into varied outcomes. In fact, some cases with isolated +8 progress quickly to AML while others remain stable for years. Some studies have analyzed the clinical variability in patients with MDS and isolated +8, precluding any clear-cut conclusions [20,21]. However, most of this research was carried out before the implementation of NGS into clinics and evidence a lack in understanding of the pathogenesis of isolated +8 patients. In view of the limited evidence on the specific impact of mutational landscape in the isolated +8 MDS setting, the present study was undertaken to determine whether the mutational profile might help to stratify patients with isolated +8. The deeply targeted gene sequencing reported in this work provides an unprecedented overview of the genomic landscape of MDS in a context of isolated +8 and provides evidence on how genetic lesions play a role in the differential outcome of MDS patients with isolated +8.

The comparison of mutational landscape between isolated +8 and non-trisomy 8 MDS showed that patients with isolated +8 displayed a differential mutational profile characterized by a slight increase in the mean number of mutations per patient (Figure 1c). In addition, mutations in *STAG2*, *RUNX1*, *EZH2*, *ZRSR2* and *IDH2* were more frequent in the isolated +8 MDS than in the control group (Figure 1b). Remarkably, the proportion of patients with isolated +8 MDS and mutations in *STAG2* was nearly three times more than in the control MDS group as previously reported [14]. As expected, the incidence of mutations in *SF3B1* was lower in the isolated +8 cohort compared to the control MDS group due to these mutations being extensively associated with MDS with ring sideroblasts [22,23,24] or normal karyotypes [25], which is concordant to the low incidence of these alterations in our cohort. By contrast, our results did not show *TP53* as frequently mutated in MDS with isolated +8, which was consistent with the high association of mutations in this gene to both complex karyotype and 5q deletion previously described in MDS [26,27,28]. All this data together could indicate a specific spectrum of genetic lesions associated with isolated +8.

The presence of *STAG2*, *SRSF2* and *RUNX1* mutations, alone or in combination, shortening both the time to leukemia transformation and the OS within MDS patients with isolated +8. Consequently, the presence of mutations in *STAG2*, *SRSF2* and/or *RUNX1* distinguishes between two different prognostic subgroups displaying specific clinical characteristics and outcomes similar to high-risk (with at least one mutation in these genes) and low-risk (without any of these genes altered) MDS patients without +8. Despite the well-known key prognostic significance that percentage of BM blasts plays in MDS, patients with isolated +8 and mutations in *STAG2*, *SRSF2* and/or *RUNX1* showed similar outcomes to high/very high-risk control patients, even with significant lower levels of blasts in BM (Appendix A). By contrast, higher levels of BM blasts were observed in isolated +8 MDS without these mutations, when compared to low-risk MDS controls. In addition, the low frequency of *SF3B1* mutations in isolated +8 patients might explain the lower levels of ring sideroblasts and platelets (of which high levels have been associated with *SF3B1* mutants [20]) observed in these patients (Appendix A). All these data suggest that not clinical, but genetic information might have an essential role in the prognosis of patients with isolated +8.

In fact, it is not the number of mutations but the number of alterations in these genes that has become an independent prognostic factor within isolated +8 MDS patients with a low percentage of blasts, as shown in the multivariate analysis in both the discovery and the validation cohorts (Table 2, Appendix A). Otherwise, when patients showed a higher percentage of blasts (>10%), as described, this clinical feature also plays a role in the prognosis of MDS [21], together with mutational status of *STAG2*, *SRSF2* and/or *RUNX1* in the specific context of isolated +8 (Appendix A).

When IPSS-R was applied to our cohort of MDS with isolated +8, 23.5% of patients included in the low-risk category and 52.4% of intermediate-risk cases categorized by the IPSS-R, showed worse outcomes and were described as high-risk isolated +8 patients due to the presence of mutations in *STAG2*, *SRSF2* and/or *RUNX1* (Figure 3a). In addition, 30.4% of patients categorized as high and very high risk by the IPSS-R were considered as low risk by the mutational profile. Therefore, our results suggest that comprehensive genomic analyses might refine the risk classification in a subset of MDS patients with isolated +8.

Otherwise, these isolated +8 differentiated subgroups were validated by the new IPSS-M classification, as a high number of patients (88.9%) with mutations in *STAG2*, *SRSF2* and/or *RUNX1* were classified as moderate–high, high and very high risk by the IPSS-M (Figure 3b). Of note, 37.5% of patients with moderate–low risk based on the IPSS-M showed a high risk based on the status of *STAG2*, *SRSF2* and *RUNX1*. In this regard, while *SRSF2* and *RUNX1* are genes considered as relevant factors increasing the score, *STAG2* shows less relevance in the IPSS-M [11]. However, *STAG2* is the second most frequent mutated gene in isolated +8 MDS patients and plays a role in disease progression and OS [19,29], as has been demonstrated in this work. Therefore, the score associated with each somatic mutation might vary depending on the subset of MDS patients studied. Specifically, analysis of mutations in *STAG2*, *SRSF2* and/or *RUNX1*, in addition to the IPSS-M, could refine prognostic information of patients presenting isolated +8. In addition, our results could be of special interest in patients classified as moderate–low risk by the IPSS-M, where *STAG2* mutations might be worsening the prognosis.

It is well known that the heterogeneity of hematological diseases such as MDS makes the management of patients difficult. Indeed, only high-risk MDS patients are eligible to receive HSCT or be treated with HMA agents due to the increasing risk of leukemia transformation [30]. The identification of patients likely to be treated in the earliest stages of the disease would enable the anticipation of clinical deterioration in patients and increase the options for stabilizing the disease when the patient’s health is in a good state. In fact, both a shorter time to initiating HMA treatment and immediate HSCT at the time of diagnosis maximized OS and complete responses in high-risk MDS patients [31,32,33], while delays during the first cycles of therapy adversely affect OS, independently of the IPSS-R status of the patient [34]. Consequently, identifying a subgroup of isolated +8 MDS with higher risk could transform the therapeutic approach of these patients.

In summary, the identification of somatic mutations has shown to be useful in distinguishing different risks within the isolated +8 MDS patients. In fact, the presence of mutations in three genes (*STAG2*, *SRSF2* and *RUNX1*) would be enough to define the course of the disease in terms of OS and AML progression. This study validates the prognosis value of the IPSS-M in the specific isolated +8 MDS subgroup, adding prognostic value to *STAG2*. Overall, these data could lead to new opportunities in the management of a high-risk subset of patients with isolated +8 MDS.

## 5. Conclusions

The mutational profile may help to correctly stratify patients with MDS and isolated +8 when clinical features identify them as intermediate risk. Indeed, mutations in *STAG2*, *SRSF2* and/or *RUNX1* distinguish a high-risk subgroup of patients with isolated +8 in terms of OS and AML transformation.

In view of these results and given the high heterogeneity of MDS, an independent mutational evaluation of each distinct biological subgroup may be helpful for a better understanding of the outcome of the disease.

## Figures and Tables

**Figure 1 cancers-15-03822-f001:**
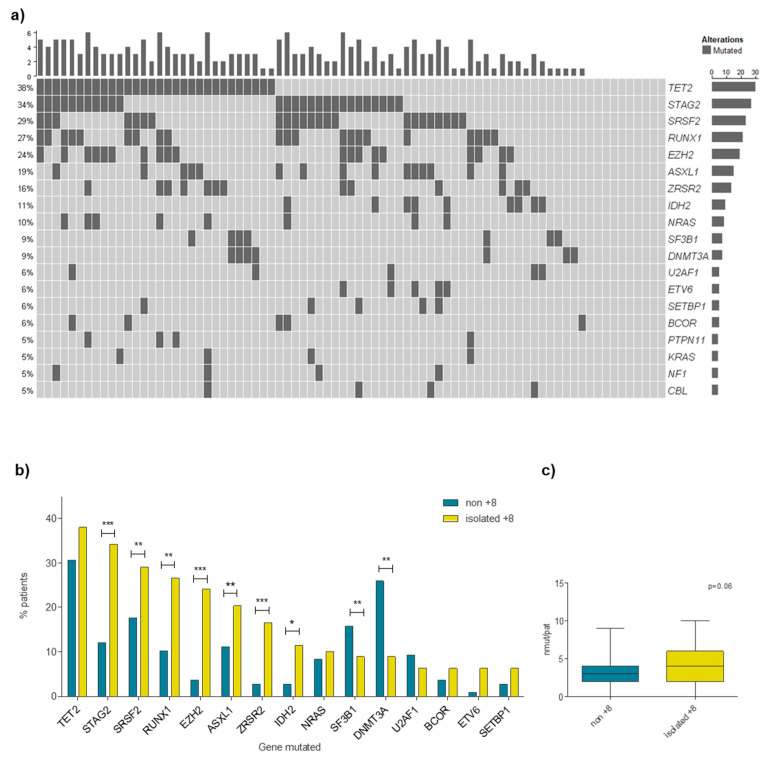
Mutational pattern of patients with MDS and isolated +8. (**a**) Mutational profile of patients with MDS and isolated +8 MDS of most frequently mutated genes (>5%); (**b**) comparison of top 15 most frequently mutated genes in isolated +8 and the control cohort (non-trisomy 8 MDS); and (**c**) mean number of mutations per patient in isolated +8 compared to the non-trisomy 8 MDS group. * *p* < 0.05; ** *p* < 0.01; *** *p* < 0.001.

**Figure 2 cancers-15-03822-f002:**
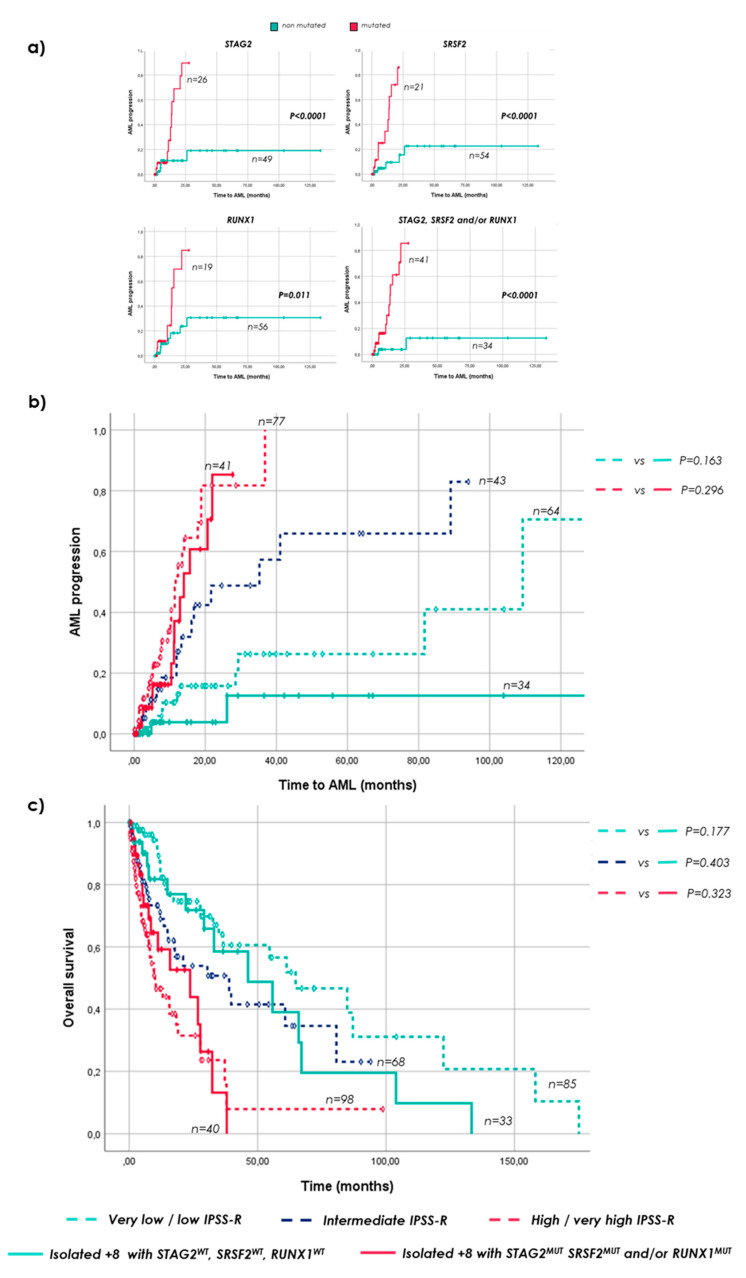
Time to AML progression and OS of insolated +8 MDS patients regarding mutational status of *STAG2, SRSF2* and/or *RUNX1*. (**a**) Time to AML progression in patients with MDS and isolated +8 according to mutational status of *STAG2*, *SRSF2* and *RUNX1*; (**b**) time to AML progression in newly described isolated +8 categories and the MDS control group stratified by IPSS-R; and (**c**) OS in newly defined isolated +8 risk categories and the MDS control cohort stratified by IPSS-R.

**Figure 3 cancers-15-03822-f003:**
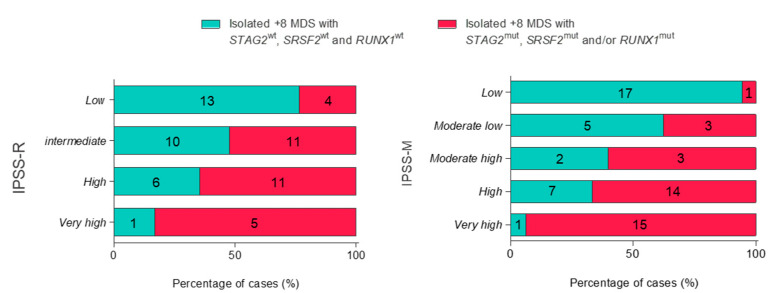
Stacked bar plots of IPSS-R and IPSS-M reclassification of isolated +8 MDS patients regarding *STAG2*, *SRSF2* and/or *RUNX1* mutational status.

**Table 1 cancers-15-03822-t001:** Clinical and biological features of isolated +8 MDS cohort.

	MDS with Isolated +8 (*n* = 94)
**Demographics**	
Age at diagnosis, years (range)	75 (33–94)
Gender, n (%)	
Female	30 (31.9%)
Male	64 (68.1%)
**MDS diagnosis and risk stratification**	
2017 WHO classification, n (%)	
MDS-SLD	3 (3.3%)
MDS-MLD	29 (31.5%)
MDS-RS	11 (12%)
MDS-del(5q)	0 (0%)
MDS-EB-1	27 (29.3%)
MDS-EB-2	19 (20.6%)
Hypoplastic MDS	3 (3.3%)
Cytogenetic risk, n (%)	
Intermediate	94 (100%)
IPSS-R category, n (%)	
Low	24 (32.9%)
Intermediate 1–2	21 (28.8%)
High	21 (28.8%)
Very high	7 (9.6%)
**Blood and bone marrow counts at diagnosis**	
Hemoglobin, g/dL (range)	9.9 (5.4–14.6)
Platelets	95 (2–520)
Neutrophils	1.7 (0.2–11.8)
Bone marrow blasts, %	4 (0–16)
Circulating blasts, %	0 (0–13)
Ring sideroblasts, %	0 (0–72)
**NGS data, n (%)**	79 (84%)

MDS-SLD: MDS with single lineage dysplasia; MDS-MLD: MDS with multilineage dysplasia; MDS-RS: MDS with ring sideroblasts; MDS-del(5q): MDS with deletion of 5q; MDS-EB-1: MDS with excess of blasts type 1; MDS-EB-2: MDS with excess of blasts type 2.

**Table 2 cancers-15-03822-t002:** Multivariate analysis of time to AML progression and OS in isolated +8 MDS patients including *STAG2*, *SRSF2* and RUNX1 mutations.

AML Progression		
Variable	Hazard Ratio (Range)	*p* Value
**Presence of mutations** (*STAG2*, *SRSF2* and/or *RUNX1*)	15.46 (2.7–88.7)	**0.002** **
**Age** (≥75 years)	0.28 (0.07–1.1)	0.075
**BM blasts** (≥5%)	0.75 (0.2–3)	0.685
**Number of cytopenias** (>2)	1.72 (0.4–7.3)	0.462
**OS**		
**Variable**	**Hazard ratio (range)**	***p* value**
**Presence of mutations** (*STAG2*, *SRSF2* and/or *RUNX1*)	3.1 (1.3–7.7)	**0.012** *
**Age** (≥75 years)	1.6 (0.7–3.6)	0.243
**BM blasts** (≥5%)	1.3 (0.5–3.6)	0.586
**Number of cytopenias** (>2)	1.2 (0.4–3.1)	0.743

* *p* < 0.05; ** *p* < 0.01.

## Data Availability

The data presented in this study are available on request from the corresponding author. The data are not publicly available due to privacy and ethical restrictions.

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
