# Peer review of "Mutational Profile Enables the Identification of a High-Risk Subgroup in Myelodysplastic Syndromes with Isolated Trisomy 8"

_cancers, 2023, doi:10.3390/cancers15153822_

Round 1

Reviewer 1 Report

Overall a very well written manuscript showing the impact of specific mutations on survival of trisomy 8 patients.   This is novel and could have an impact on treatment and prognosis of these patients.  

Only a minor amendment would be to add into Figure 2a another figure showing time to AML with any of the three specified mutations.

Reviewer 2 Report

In the manuscript, the authors performed targeted-deep sequencing analysis of 79 cases of MDS with isolated trisomy 8 (+8) and identified STAG2, SRSF2 and/or RUNX1 mutations as the independent prognostic indicators for a high-risk subgroup of MDS w/ isolated +8. This manuscript may provide some useful information, although the prognostic significance of STAG2, SRSF2 and RUNX1 mutations in MDS is well-studied, and a recent study also identified EZH2, ASXL1 and STAG2 as independent prognostic factors in MDS with isolated +8 (Drevon et al. Br J Haematol. 2018). The following are my major concerns about this manuscript.

1.     In the Introduction (lines 80-84), the authors stated “In that way, some studies have described the presence of mutated genes in MDS displaying +8, such as ASXL1, EZH2, STAG2, SRSF2 or U2AF1[14–17]. However, the role of somatic mutations is still controversial as there is no evidence of its impact in the outcome of isolated in large cohorts of MDS patients”. I don’t think that this statement is correct because the previous study by Drevon et al. (Br J Haematol. 2018), which is larger than the current study, identified the mutations of EZH2, ASXL1, and STAG2 as poor prognostic markers in MDS w/ isolated +8.

2.     I also don’t understand why the authors did not identify EZH2 and ASXL1 mutations as poor prognostic markers in this study as shown by the previous study. Any possible explanations?

3.      Regarding Figure 1, there seems a very significance in SRSF2, ASXL1 and DNMT3A mutations between non +8 and isolated +8 cases (to my eyes). However, the authors did not mark any of those genes with *p<0.05; **p<0.01; ***p<0.001. Is there significant difference? Why were there no SD values associated with those bars?

4.     Were those mutations present alone or concurrent with other mutations, i.e., co-mutations, in their study?

5.     Regarding Table 2, the authors stated “despite having higher hemoglobin levels and lower counts of blast in BM, no differences were found in patient’s outcome in terms of both AML progression and OS”, which is counter-intuitive. Any possible explanation for that? How were the blasts counted by manual diff on aspirate smears, flow cytometry or IHC?

6.     Regarding Figure 3, it showed that age, BM blasts and cytopenia(s) had no significant impact on AML progression in their cases. This does not make sense to me because the prognostic significance of these factors has been well-established. This also seem contradictory to their own figure 4 data. I don’t not understand their data and figures here.

Reviewer 3 Report

In this study, Toribio-Castello et al analyze the molecular panels of MDS patients with isolated +8 by cytogenetics and report adverse prognosis associated with STAG2, SRSF2 and RUNX1 mutations. The study adds value by analyzing a relatively large cohort MDS with +8, which accounts for ~5-10% of MDS cases. The result section provides insight into the molecular landscape of MDS with +8. However, the implications of these findings are overestimated and I worry about several statements that cannot be derived from these results. In general, this manuscript is twice as long as it should be and needs major revisions. The quality of writing and structure of the methods are limited. I have the following concerns with this report:

1. In the summary and discussion, the statement that the study findings can help identify a high risk subgroup should be removed. The patients in this report would already be considered high risk based on molecular mutations in STAG2, SRSF2 and RUNX1, all of which have been reported as prognostically adverse before. This is not a new finding and does not identify a different subgroup. This work only confirms that the prognostic value of the named mutations applies in the particular group of MDS patients they are studying.

2. In the summary and discussion, the statement about identification of new drug targets should also be removed. Indeed, one can argue that finding targeted agents for STAG2, SRSF2 and RUNX1 would be personalized medicine, but that would have nothing to do with +8 and any MDS patient can benefit.

3. In introduction, the wording needs to be changed. The authors underestimate the value of molecular mutations by saying: “risk stratification remains heavily reliant on cytogenetics”, which is not true in 2023 and cannot be sustained by citing a paper from 2012. Though it is true that IPSS-M has been recently developed, most clinicians have used molecular abnormalities for several years at this point. The statement “the role of somatic mutations is still controversial in MDS with +8” should also be revised. Next gen sequencing of MDS patients have been part of clinical practice for years.

4. The methods do not include the time period in which the cohort was identified. Why do only 74 of 94 patients had molecular testing? Is this an old cohort? Is that why 2017 categories are reported?

5. The methods described are confusing, is the comparison between matched controls or to an external cohort or both? Why do the authors compare 94 patients with +8 if only 74 can be assessed for the predictor variable (mutations?). Why do the authors say they sequenced 187 patients? Where did this number come from?

6. Remove detailed descriptions of cytogenetics and FISH techniques from the methods. This is not relevant information. It seems that the authors utilized a home NGS panel of 92 genes which can be mentioned in summary, but the readers do not need such a detailed description of NGS sequencing in a clinical paper.

7. Please provide a justification for censoring at time of HSCT in overall survival analysis. This is not common practice.

8. The authors should provide a justification for why they decided to compare the subsets of +8 MDS with and without mutations to high or low IPSS risk groups. One is a molecular risk stratifying system, the other one is driven by clinical characteristics. Where did the patients without +8 come from? In methods, the authors report 2 comparison groups and the reader cannot tell which one they are using. These issues need to be included in the methods and the paragraph in page 9 (lines 239 – 242) belongs in the discussion section, not results. I have my reservations with these comparisons. Are these patients matched by IPSS-R? I can only assume they are not since there are significant clinical differences. In which case, the authors are comparing 2 very different clinical populations.

9. Figure 3 can be a table or text, it is unnecessary to present HRs as a graph

10. The results on “misclassification” by IPSS-R are a consequence of molecular data adding value for risk stratification. This has been documented across all subsets of MDS in various previous studies. There is value in this report of +8 MDS patients as confirmatory findings, but it is not surprising that IPSS-M better risk stratifies MDS. Results in page 10 needs to be summarized and figure 4 needs to mention the number of patients in each subgroup.

In this study, Toribio-Castello et al analyze the molecular panels of MDS patients with isolated +8 by cytogenetics and report adverse prognosis associated with STAG2, SRSF2 and RUNX1 mutations. The study adds value by analyzing a relatively large cohort MDS with +8, which accounts for ~5-10% of MDS cases. The result section provides insight into the molecular landscape of MDS with +8. However, the implications of these findings are overestimated and I worry about several statements that cannot be derived from these results. In general, this manuscript is twice as long as it should be and needs major revisions. The quality of writing and structure of the methods are limited. I have the following concerns with this report:

1. In the summary and discussion, the statement that the study findings can help identify a high risk subgroup should be removed. The patients in this report would already be considered high risk based on molecular mutations in STAG2, SRSF2 and RUNX1, all of which have been reported as prognostically adverse before. This is not a new finding and does not identify a different subgroup. This work only confirms that the prognostic value of the named mutations applies in the particular group of MDS patients they are studying.

2. In the summary and discussion, the statement about identification of new drug targets should also be removed. Indeed, one can argue that finding targeted agents for STAG2, SRSF2 and RUNX1 would be personalized medicine, but that would have nothing to do with +8 and any MDS patient can benefit.

3. In introduction, the wording needs to be changed. The authors underestimate the value of molecular mutations by saying: “risk stratification remains heavily reliant on cytogenetics”, which is not true in 2023 and cannot be sustained by citing a paper from 2012. Though it is true that IPSS-M has been recently developed, most clinicians have used molecular abnormalities for several years at this point. The statement “the role of somatic mutations is still controversial in MDS with +8” should also be revised. Next gen sequencing of MDS patients have been part of clinical practice for years.

4. The methods do not include the time period in which the cohort was identified. Why do only 74 of 94 patients had molecular testing? Is this an old cohort? Is that why 2017 categories are reported?

5. The methods described are confusing, is the comparison between matched controls or to an external cohort or both? Why do the authors compare 94 patients with +8 if only 74 can be assessed for the predictor variable (mutations?). Why do the authors say they sequenced 187 patients? Where did this number come from?

6. Remove detailed descriptions of cytogenetics and FISH techniques from the methods. This is not relevant information. It seems that the authors utilized a home NGS panel of 92 genes which can be mentioned in summary, but the readers do not need such a detailed description of NGS sequencing in a clinical paper.

7. Please provide a justification for censoring at time of HSCT in overall survival analysis. This is not common practice.

8. The authors should provide a justification for why they decided to compare the subsets of +8 MDS with and without mutations to high or low IPSS risk groups. One is a molecular risk stratifying system, the other one is driven by clinical characteristics. Where did the patients without +8 come from? In methods, the authors report 2 comparison groups and the reader cannot tell which one they are using. These issues need to be included in the methods and the paragraph in page 9 (lines 239 – 242) belongs in the discussion section, not results. I have my reservations with these comparisons. Are these patients matched by IPSS-R? I can only assume they are not since there are significant clinical differences. In which case, the authors are comparing 2 very different clinical populations.

9. Figure 3 can be a table or text, it is unnecessary to present HRs as a graph

10. The results on “misclassification” by IPSS-R are a consequence of molecular data adding value for risk stratification. This has been documented across all subsets of MDS in various previous studies. There is value in this report of +8 MDS patients as confirmatory findings, but it is not surprising that IPSS-M better risk stratifies MDS. Results in page 10 needs to be summarized and figure 4 needs to mention the number of patients in each subgroup.

Round 2

Reviewer 2 Report

I appreciate that the authors made efforts to address my concerns. However, it seems that the authors did not understand my concern about their data and statement regarding the prognostic value of the blasts or cytopenia(s).

Let me make myself clear about my concern and why this is important. I believe that the presence of STAG2, SRSF2 and/or RUNX1 mutations is an independent prognostic factor for adverse outcome as it has been demonstrated by other studies in various MDS and AML subtypes with or without +8. This is not an issue to me at all. My big concern is about their data (the new Table 3), which show no prognostic significance of the blasts and cytopenia(s) in their cases, and their conclusion about the prognostic insignificance of blasts and cytopenia(s) based on their data, which is likely to be wrong and misleading (in my opinion) because, as we all now, blasts and cytopenia(s) are well-established independent prognostic parameters in MDS. This has been proven by  hundreds of large size studies. As matter of fact, the data from their Tables 3 and 4 also confirm the importance of blasts and cytopenia(s) in their cases.

The authors mentioned “the absence of these mutations might improve outcome in this isolated +8 subset of MDS patients, even when blast percentages are higher than in low risk patients (3% in isolated +8 MDS without these mutations vs 1% in the low risk MDS control cohort)”. This is not nearly enough to disprove the prognostic importance of blasts and cytopenia(s) in their cases because (to me) there is no meaningful difference between 3% vs. 1% blasts, which is within the range of individual variances. To prove their argument (of prognostic unimportance of blasts and cytopenia(s) in their cases), the authors need to compare OS and AML progression in patients with different ranges of blasts and cytopenia(s), for example, 1-4% vs. 5-9% vs. 10-19% blasts and <500/uL vs. 500-1,000/uL, vs. >1,000/uL ANC, with and without +8, in their case. I expect that the authors will find blasts and cytopenia(s) to be independent prognostic factors, just like the presence of STAG2, SRSF2 and/or RUNX1 mutations, in their cases. I will be happy if they can prove me wrong on this as well.

Reviewer 3 Report

The authors provided satisfactory responses to most of my concerns and have prepared an improved manuscript in my opinion. I commend the authors for this work. However, I respectfully would like to point out 2 major issues I am still worried about:

1. I asked for a justification about censoring outcomes at the time of HSCT. Though I agree about HSCT influencing survival, one cannot assume end of follow up at time of HSCT without introducing bias. As examples, this strategy can a) misclassify survival of higher risk cases as shorter because these individuals are the most likely transplanted quickly, or b) shorten follow up time to identify AML transformation (which can occur following MDS relapse post HSCT). How do we know that this censoring is not the cause of increased blasts/cytopenias not being linked to AML or OS in what it is now table 4? Another reviewer pointed out that these are established factors and questioned the finding. I strongly recommend using the standard definition of OS. The authors mention no relapses were identified so perhaps example “b” is not present but I am still concerned about survival of many IPSS-R high/very-high +8 MDS cases in this study being misclassified. How many patients underwent HSCT? Based on strong effect sizes in multivariable regression, I cannot imagine the primary predictor will be different if OS definition is changed, but the statistics would be more reliable. If this is not done, I suggest including a reference to previous studies that have censored at time of HSCT because it is not standard practice.

2. I pointed out that tables 2 and 3 are comparisons of very different groups. Based on the authors reply, I still think these comparisons do not add much value. I think the authors aim to communicate that STAG2/SRSF2/RUNX1 convey prognostic information. For example, they mention that outcomes of mutation-positive +8 cases were similar to high risk cases without +8 despite lower blasts. This makes me think they want to underscore the prognostic significance of the mutations despite blast number. This is achieved by adjusting for blasts in a multivariable model of +8 cases (which they do in Table 4), not by comparing to a completely different group of MDS without +8. In my opinion, the message of the study is well documented in figure 2 (survival models), table 4 (multivariable regression), and figure 3 (redefinition of risk). The comparison to cases without +8 is redundant for the paper’s message and adds unnecessary complexity. I would move it to supplemental data.

Round 3

Reviewer 3 Report

I appreciate the author's effort to address my concern. I believe the current version is significantly improved and ready for publication.